# Low doses of diarrhoeagenic *E. coli* induce enhanced monocyte and mDC responses and prevent development of symptoms after homologous rechallenge

**Mojtaba Porbahaie**[1]*, **Maartje van den Belt**[2], **Laurien Ulfman**[3], **Rianne M. A. J. Ruijschop**[2], **Elly Lucas–van de Bos**[2], **Anita Hartog**[2], **Stefanie Lenz**[2], **Ingrid J. van Alen-Boerrigter**[2], **Malgorzata Teodorowicz**[1], **Huub F. J. Savelkoul**[1], **Wim Calame**[4], **Els van Hoffen**[2], **R. J. Joost van Neerven**[1,3]☯*, **Alwine Kardinaal**[2]☯

1 Cell Biology and Immunology Group, Wageningen University & Research, Wageningen, The Netherlands, 2 NIZO Food Research, Ede, The Netherlands, 3 FrieslandCampina, Amersfoort, The Netherlands, 4 StatistiCal, Wassenaar, The Netherlands

☯ These authors contributed equally to this work.
* mojtaba.porbahaie@wur.nl (MP); joost.vanneerven@wur.nl (RJJN)

**Data Availability Statement:** All relevant data are within the paper and its Supporting information files.

## Abstract

The experimental challenge with attenuated enterotoxigenic *E. coli* strain E1392/75-2A prevents diarrhea upon a secondary challenge with the same bacteria. A dose-response pilot study was performed to investigate which immunological factors are associated with this protection. Healthy subjects were inoculated with increasing *E. coli* doses of 1E6-1E10 CFU, and three weeks later, all participants were rechallenged with the highest dose (1E10 CFU). Gastrointestinal discomfort symptoms were recorded, and stool and blood samples were analyzed. After the primary challenge, stool frequency, diarrhea symptom scores, and *E. coli*-specific serum IgG (IgG-CFA/II) titer increased in a dose-dependent manner. Fecal calprotectin and serum IgG-CFA/II response after primary challenge were delayed in the lower dose groups. Even though stool frequency after the secondary challenge was inversely related to the primary inoculation dose, all *E. coli* doses protected against clinical symptoms upon rechallenge. *Ex vivo* stimulation of PBMCs with *E. coli* just before the second challenge resulted in increased numbers of IL-6$^+$/TNF-α$^+$ monocytes and mDCs than before the primary challenge, without dose-dependency. These data demonstrate that primary *E. coli* infection with as few as 1E6 CFU protects against a high-dose secondary challenge with a homologous attenuated strain. Increased serum IgG-CFA/II levels and *E. coli*-induced mDC and monocyte responses after primary challenge suggest that protection against secondary *E. coli* challenges is associated with adaptive as well as innate immune responses.

**Funding:** The MIRRE study was one of the intervention studies performed by the CHALLENGE consortium. This consortium is financially supported by 'Europees fonds voor regionale ontwikkeling (EFRO)' Operationeel Programma Oost-Nederland (EFRO OP Oost 2014-2020 CHALLENGE PROJ_00676) and co-funded by NIZO, FrieslandCampina and Nutrileads. The salaries of FrieslandCampina employed authors [RJJvN and LU] were part of the cofinancing of the study. These authors contributed to the design of the study, as did all authors identified in the 'author contributions' section. FrieslandCampina did not have any additional role in the study design, data collection and analysis, decision to publish, or preparation of the manuscript. The specific roles of these authors are articulated in the 'author contributions' section.

**Competing interests:** L. Ulfman and R.J.J. van Neerven are employed by FrieslandCampina, and MP was supported by a grant from FrieslandCampina. The MIRRE study was a model development study and no commercial or other dairy products were testd, so the study was not directly related to the business of FrieslandCampina, other than being part of the CHALLENGE consortium, in which dairy products were tested in another study. Other authors declare that there is no conflict of interest. This does not alter our adherence to PLOS ONE policies on sharing data and materials.

## Introduction

Diarrhea is prevalent worldwide and is one of the leading causes of death in all age groups, with children under five being the most impacted [1]. Foodborne diarrhea primarily affects inhabitants of developing countries with low sanitary standards and also affects visitors to these endemic areas, resulting in travelers' diarrhea. Etiologically, diarrheagenic *Escherichia coli* (DEC) -and most prominently enterotoxigenic *E. coli* (ETEC)- is the primary cause of these diarrheal diseases [2–4]. These ETEC strains can survive gastrointestinal digestion and transit successfully via the gastrointestinal tract to enter the intestine [5, 6]. These bacteria adhere to the intestine's gut mucosa, where the secreted toxins bind to their respective host cell receptors and eventually deregulate ion channels and induce diarrhea [3, 7, 8]. Due to the high mortality rates in children and the consequences in adults, including (temporary) impairment, ETEC infections have become a global health burden that must be addressed.

In the absence of a broadly effective approved vaccine against ETEC, improving host resistance to infection through dietary or pharmacological intervention may be a viable alternative strategy for reducing diarrhea. It has been demonstrated that dietary supplementation with calcium in the form of dairy products [9], calcium in combination with probiotics [10], and a combination of zinc and iron [11] reduce the occurrence and severity of diarrhea. Dietary components may prevent the pathogen from colonizing the gut [9], act as bactericidal agents [12], and/or enhance host immune responses. These examples imply the efficacy and benefit of dietary prevention strategies.

To evaluate the clinical efficacy of drugs or food ingredients in preventing diarrhea, infection challenge models have been employed in which healthy volunteers are challenged with (attenuated) live ETEC [12, 13]. One of the established models uses attenuated ETEC strain E1392/75-2A [9, 14, 15]. It is worth mentioning that although the strain applied in those studies does not produce enterotoxins, it induces diarrhea, making it a valuable model for studying mechanisms. It was shown that protection against ETEC strain E1392/75-2A provides approximately 75% protection against virulent enterotoxin-producing strains [16]. Therefore, this article refers to this strain as diarrheagenic *E. coli* (dia. *E. coli*) due to no enterotoxin production.

A recent study has addressed that a primary dia. *E. coli* inoculation protected the subjects against reinfection [15]. In this study, the primary challenge with a high dose of dia. *E. coli* strain E1392/75-2A (O6:H16) resulted in protection against the second challenge with the same bacteria three weeks later. However, it is unclear which immunological mechanisms confer protection and whether a low dose primary challenge results in decreased or absent protection. If this is the case, it provides a valid model for studying the impact of drug and food components on long-term protection against *E. coli* infection and enables us to investigate the protection mechanisms following the second exposure.

To this aim, we conducted a dose-response pilot study using the dia. *E. coli* strain E1392/75-2A challenge model. The primary challenge in this model was performed with bacteria at doses ranging from 1E6 to 1E10 Colony-Forming Unit (CFU), and the second challenge was performed with the standard high dia. *E. coli* dose of 1E10 CFU. First, we aimed to identify the lowest dose capable of conferring protection against the second challenge. For this reason, clinical outcomes for stool frequency, stool consistency, fecal wet weight, and scores on the Gastrointestinal Symptoms Rating Scale (GSRS) [17] were quantified to monitor diarrhea progression. Furthermore, immune responses prior to and following the primary and secondary challenges were further characterized to elucidate the immunological mechanisms behind putative protection. Induction of serum IgG against dia. *E. coli*-specific Colonization Factor Antigen II (IgG-CFA/II) was followed, calprotectin, β-defensin, and secretory IgA (SIgA)

levels in fecal water were determined, and *ex vivo* response of monocyte and dendritic cell to the *E. coli*, toll-like receptor 4 (TLR-4), and TLR-5 stimulation were evaluated.

## Methods

### Study design, participants, and specimens

The MIRRE study was designed as a randomized, double-blinded, parallel dose-response, 7-week infection challenge study in 30 healthy adult male volunteers. The selected participants' age was between 18 to 55 years, with a BMI ranging from 18.5–30 kg/m$^2$. The study was approved by the Medical Ethics Committee (METC) of Brabant (Tilburg, the Netherlands) and was conducted according to the Declaration of Helsinki and was registered at www. clinicaltrials.gov with identification number: NCT03596827 (First registration date 24/07/2018). The CONSORT flow diagram of the study can be find on Fig 1.

All participants were medically evaluated based on self-report following recruitment and provision of written informed consent. Participants were considered for participation if they met the inclusion criteria, and there were no indications of exclusion criteria (Fig 1). Subjects were stratified by age and BMI (as determined during pre-study screening) and then assigned randomly to one of five treatment groups (n = 6 per group). Stratification and randomization of study participants, and blinding and labeling of dia. *E. coli* dosages, were coordinated by a scientist not involved in the project. Strata were defined manually in MS Excel using the individual data. Stratified randomization of participants to treatment group was performed using the "Research Manager" software (Research Manager, Deventer, The Netherlands). The researchers on the project team and the study participants were kept blind to the treatment assignment.

During a two-week acclimation period (Fig 2), participants were instructed to maintain their usual physical activity pattern and habitual food intake; however, they needed to reduce and standardize their dietary calcium intake (<500 mg/day) [9]. After a standard low-calcium dinner and an overnight fast, the participants were assigned to dosage groups. On study day 14, they were orally inoculated with the live-attenuated dia. *E. coli* strain E1392/75-2A (supplier: Acambis, Cambridge, UK) [9, 18]. Each group received a different dose (1E6, 1E7, 1E8, 1E9, or 1E10 CFU), and three weeks later (day 35), all participants were given the standard dose of 1E10 CFU. Participants first received a NaHCO$_3$ solution (100 mL 2% NaHCO$_3$) to neutralize the gastric acid. After 5 minutes, they received a fruit juice (100 mL, pH 7.4) containing the attenuated dia. *E. coli* strain at the doses mentioned earlier. Participants were requested to restrict their probiotics, medication, and alcohol intake for three days before and four days after challenges and record their clinical gastrointestinal symptoms daily using an online questionnaire. Multiple blood and fecal samples were taken for analysis at various time points. (Fig 2).

### Online questionnaire

From day 11 till day 17 (interval I) and from day 32 till day 38 (interval II), participants had to report information on stool frequency (total number of stools per day) and stool consistency according to the Bristol Stool Scale (ranging from 1 as constipation to 7 as watery diarrhea). Moreover, participants were instructed to record the frequency and severity of symptoms by the validated Gastrointestinal Symptom Rating Scale (GSRS) [17]. The GSRS is a disease-specific scale of 15 questions related to five subdomains: diarrhea, abdominal pain, indigestion, reflux, and constipation. The GSRS has a seven-point graded Likert-type scale where 1 represents the absence of troublesome symptoms and 7 represents very troublesome symptoms. The later analysis of the GSRS score focused on the GSRS total daily score, subdomain diarrhea

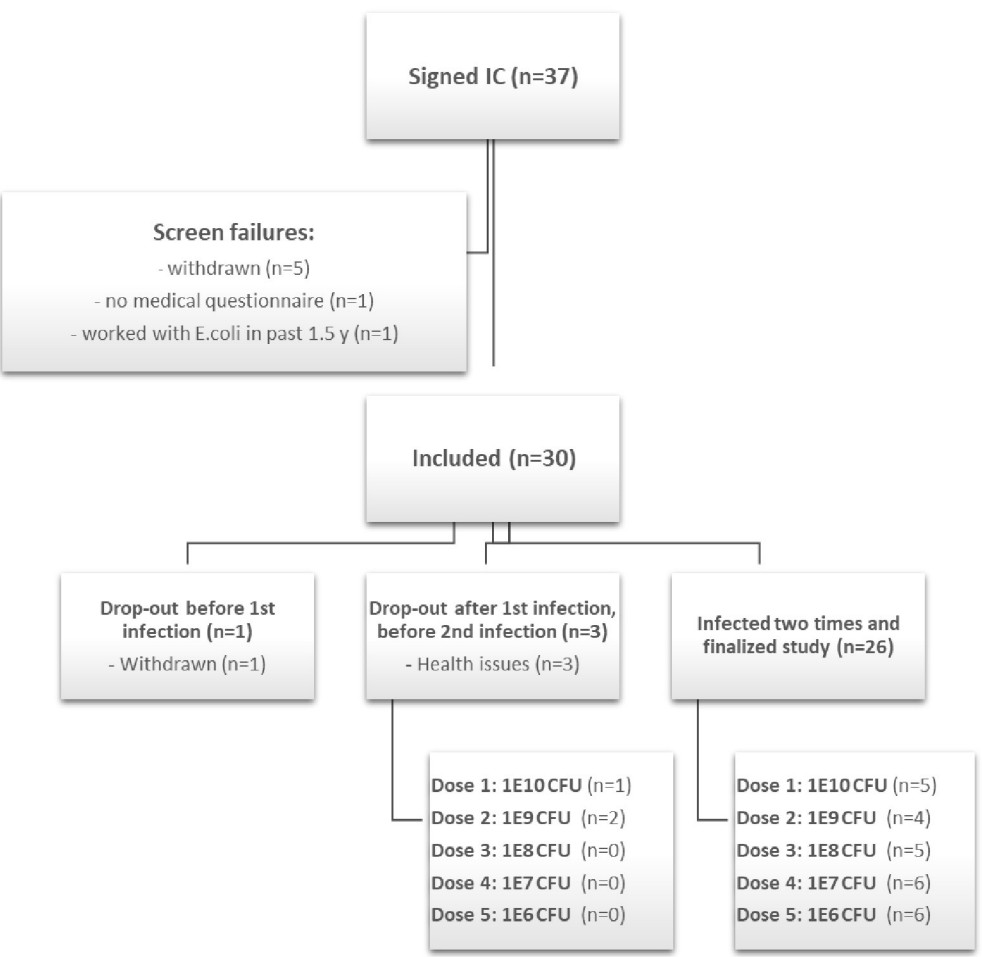

**Fig 1. CONSORT flow diagram of MIRRE study.** The participants who signed the informed consent were further assessed for inclusion and exclusion criteria. The main inclusion criteria included sex, age, BMI, health condition, and the willingness to comply with all study procedures. The exclusion criteria were: current or previous underlying gastrointestinal disease; confirmed *E. coli* or cholera infection within 3 years prior to inclusion; diarrhea symptoms with a history of travel to *E. coli* endemic regions in past 3 years; vaccination for or ingestion of *E. coli* or cholera within 3 years before inclusion; known allergy to antibiotics; reported average stool frequency of <1 or >3 per day; use of antibiotics, activated charcoal, laxatives up till 6 months before inclusion; use of gastric acid suppression medication within 3 months before inclusion; current excessive alcohol consumption or drug (ab)use; vegans. Regarding the drop-out participant, One participant was dropped out of the study before the first infection and three others have withdrawn from the study before the second infection because of health issues. From those three, one participant had respiratory tract infections (not related to the study) and the second one experienced gastrointestinal symptoms before the second infection (not related to the study). One of the dropped out participants had serious adverse event (SAE) because of hospitalization due to the risk of dehydration. This subject suffered from diarrhea, nausea, vomiting, fainting and an overall feeling of malaise one day after the *E. coli* inoculation. The SAE was reported online to the national authority (CCMO) within 7 days after the reporting to the study team. The subject was hospitalized for 7 days, because of flu-like symptoms. The medical investigator assessed that the SAE was probably related to the study intervention, and that the SAE was completely resolved after discharge from the hospital. The medical investigator also consulted the general practitioner of this subject. It appeared that there were pre-existing factors that were probably triggered by the infection, which may have contributed to the need for (prolonged) hospitalization of this subject.

(loose stools, increased passage of stools, and urgent defecation), and subdomain abdominal pain (abdominal discomfort, sucking sense epigastrium, nausea). These two subdomains were hypothesized to be affected the most by the dia. *E. coli* challenge. The GSRS total score ranges from 15–105, and the scores for the subdomains diarrhea and abdominal pain range from 3–21. The "Research Manager" program was used for data collection and management.

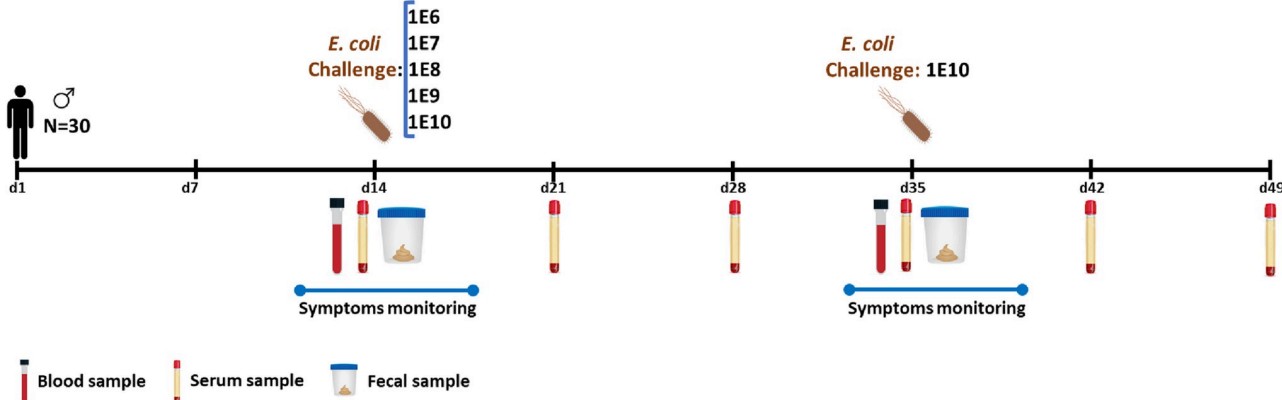

**Fig 2. A graphical representation of the MIRRE study design: 30 male participants were recruited and randomized into five separate groups.** On day 14, each group orally received different doses (1E6, 1E7, 1E8, 1E9, and 1E10 CFU) of dia. *E. coli* strain E1392/75-2A, and three weeks later on day 35, all participants were challenged with a dia. *E. coli* dose of 1E10 CFU. Three days before until four days after each challenge, participants were asked to report the clinical gastrointestinal symptoms. Blood and stool samples were collected for later analysis at multiple time points.

### Collection and characterization of fecal samples

Fecal samples were collected a day before (day 12/13; day 33/34) and three days after each challenge (days 15–17; days 36–38). Participants were asked to collect all 24h stool samples in collection bags and freeze them on-site using the provided mini freezer. The frozen samples were transported to the analysis center, sorted, weighed, homogenized, aliquoted, and stored at -20˚C until later analysis. Total fecal wet weight (24h pooled) was measured according to standard protocols (internal validated procedure), and the % fecal wet weight was quantified after the freeze-drying of the samples.

### Quantification of intestinal immune biomarkers

The amount of calprotectin, β-defensin, and total secretory IgA (SIgA) in the participants' fecal water was measured by ELISA. Fecal water preparation and subsequent detection ELISA was performed with the provided reagents and according to the manufacturer protocol (ImmunDiagnostik, Bensheim, Germany; Calprotectin:K6927, β-defensin:K6500, SIgA: K8870). Two separate dilutions of the fecal water were used to assure in-range measurements, and the results are expressed as the amount of analyte per gram of fecal dry weight.

### Measurement of IgG-CFA/II serum levels

On six timepoints during the study (days 14, 21, 28, 35, 42, and 49), blood samples were collected to measure serum specific IgG against CFA/II. For blood sample collection, Vacutainer serum tubes (Becton Dickinson 367895, Plymouth, United Kingdom) were used, and the sera were collected after centrifugation (2000xg, 10 min., RT). The centrifugation was done within 2 hours from the blood collection, and the resulting sera were stored at -80˚C. The amount of CFA/II specific IgG was quantified by ELISA as described elsewhere [9]. A CFA/II-IgG-positive serum from a previous study was used as the standard curve [15], and values are expressed as Arbitrary Units (AU)/mL.

### Activation of monocytes, mDC, and pDC by dia. *E. coli* and TLR-ligands

Just before bacteria inoculation (day 14 and day 35), blood samples from participants were collected in Vacutainer (K2-EDTA) tubes (Becton Dickinson 367525, Franklin Lakes, NJ, USA).

**Table 1. Antibodies panel used for PBMCs immunological assay.**

| Antibody | Fluorochrome | Host | Isotype | Clone | Company | Catalog number |
|---|---|---|---|---|---|---|
| α-CD3 | FITC | mouse | IgG1 | UCHT1 | Biolegend | 300406 |
| α-CD11c | BV 421 | mouse | IgG1 | 3.9 | Biolegend | 301628 |
| α-CD14 | Percp-Cy5.5 | mouse | IgG1 | HCD14 | Biolegend | 325622 |
| α-CD19 | FITC | mouse | IgG1 | SJ25C1 | Biolegend | 363008 |
| α-CD20 | FITC | mouse | IgG2b | 2H7 | Biolegend | 302304 |
| α-CD56 | FITC | mouse | IgG1 | HCD56 | Biolegend | 318304 |
| α-HLA-DR | BV 510 | mouse | IgG2a | L243 | Biolegend | 307646 |
| α-CD123 | PE-Cy7 | mouse | IgG1 | 6H6 | Biolegend | 306010 |
| α-IL-6 | PE | rat | IgG1 | MQ2-13A5 | Biolegend | 501107 |
| α-TNF-α | AF647 | mouse | IgG1 | MAb11 | Biolegend | 502916 |

Company affiliation is Biolegend (San Diego, CA, USA)

The Peripheral Blood Mononuclear Cells (PBMCs) were isolated by density gradient centrifugation using Ficoll Paque Plus (GE Healthcare 17-1440-02, Chicago, IL, USA). The isolated cells were seeded (2E6 cell/well) in 12-well plates (Costar CL3513, Sigma-Aldrich, St. Louis, MO, USA) and subsequently stimulated with either medium (RPMI-1640), 200 ng/mL of LPS (Sigma L2880, Sigma-Aldrich, St. Louis, MO, USA), 500 ng/mL of Flagellin (Invivogen tlrl-stfla, San Diego, CA, USA), or 1E7 CFU/well dia. *E. coli* (strain E1392/75-2A). Brefeldin A (BFA) (Invitrogen 00-4506-51, Carlsbad, CA, USA) was added to the wells to keep the produced cytokines inside the cells, and the plates were incubated for 3 hours at 37°C with 5% of $CO_2$. After the incubation, the cells were harvested and stained with fluorochrome-conjugated antibodies against extracellular markers for cell phenotyping (Table 1). Next, Fixable Viability Dye eFluor 520 (eBioscience 65-0867-14, San Diego, CA, USA) was applied as the live-dead marker. After cell fixation and membrane permeabilization, the production of IL-6 and TNF-α was determined intracellularly by staining the cells with flow cytometry antibodies (Table 1). The samples were measured on BD FACS CANTO II, and the generated flow cytometry data were analyzed with FlowJo v10 (FlowJo LLC, Ashland, OR, USA). The gating strategy to identify cells is shown on (S1 Fig).

## Power calculation and statistical analysis

This research was designed as a pilot study. Since no information about the impact by bacterial doses was available, a specific setup, a so-called Design of Experiment (DOE: Plackett-Burman design [19]) with minimum and maximum challenge levels, was designed. Several doses were included with a calculated number of subjects in each group (n = 6). By applying this DOE approach, it was expected to identify the dose-dependent trends and potentially a single dose optimum. "In short, the highest and the lowest dose of infectious E. coli were used as two factors in a 6-run approach while the other 3 doses were used to analyze the overall variance in the outcome of the experiment. Since this process was repeated 5 times (with the 5 doses applied in the study, the combination of two other doses as explanatory factor and the remaining ones as variance building blocks), a total number of 30 persons was included in the current experiment.

Correlation between parameters was investigated via General Estimating Equations (GEE) modelling including confounding parameters in a stepwise approach. When identified, a graphical check with the two parameters was done to verify their relationship. In the GEE model, the dependent parameter, such as serum IgG levels of CFA/II, was associated with various independent parameters, such as day and dose, as well as confounding factors, such as

BMI, age, and start value. Moreover, non-linear functionalities of day and dose have been included to correct for these types of association. Fit of the model was assessed via Wald Chi-square. Only when the fit was significant the impact by the various independent parameters on the dependent parameter was confirmed. One-way ANOVA post-test was applied to check the linear trends. For statistical analysis of the PBMCs work, paired sample t-test was applied to compare the outcomes of d14 vs. d35 within each stimulation. The differences were considered significant for the whole dataset when the p-value was <0.05.

## Results

### Baseline characteristics of the participants

The baseline characteristics of the participants are summarized in Table 2. Healthy male participants (n = 30) were randomized and divided into 5 study groups receiving different doses of dia. *E. coli* strain E1392/75-2A. Each group consisted of 6 participants; however, four individuals had withdrawn from the study at different time points indicated in Table 2 and described in Fig 1.

### Gastrointestinal symptom rating scale (GSRS)

The GSRS score was analyzed as the total daily score as well as for the subdomain diarrhea and subdomain abdominal pain at two time periods before and after each dia. *E. coli* inoculation. These analyses were done for interval I (days 12/13-17) and interval II (days 33/34-38).

After the primary challenge (interval I), GSRS domain diarrhea scores increased in a significant dose-dependent manner (P <0.05), with the score peak on day 15 (Fig 3A). GSRS domain abdominal pain (Fig 3B) and GSRS total daily score (Fig 3C) showed similar trends in the response; however, no significant dose-dependency was noted. During interval II, no major variation in any GSRS scores was recorded for either of the primary doses compared to before the second challenge (Fig 3A–3C).

### Stool parameters

Stool parameters, including total and percentage fecal wet weight (%WW), stool frequency, and stool consistency, were determined for interval I (days 12/13-17) and interval II (days 33/34-38) during the study. An overall increase in total fecal wet weight and %WW was observed in interval I starting from day 15 for doses 1E9 and 1E10, where the increase was delayed for other doses by one day (Fig 4A and 4B). No significant dose-dependency in the responses was present, and all the groups returned to the baseline levels by day 17. In interval II, total fecal wet weight and %WW either showed no variations or decreased with no dose-dependency, and the highest levels did not surpass the maximum values of interval I. The baseline %WW values on days 13 and 34 were deducted from participants' maximum values during intervals I and II, respectively, and presented as delta % fecal wet weight (Δ%WW). This parameter

**Table 2. Baseline characteristics of study participants.**

| Variable | | Dose 1: | Dose 2: | Dose 3: | Dose 4: | Dose 5: |
|---|---|---|---|---|---|---|
| Dose | | 1E10 | 1E9 | 1E8 | 1E7 | 1E6 |
| Number of participants | | 6 (1*) | 6 (2*) | 6 (1*) | 6 | 6 |
| Age | Mearefen (SD) | 35.8 (15.2) | 33.7 (10.6) | 35.2 (13.1) | 40.8 (7.7) | 37.5 (10.9) |
| BMI (kg/m$^2$) | Mean (SD) | 22.9 (2.8) | 23.8 (2.6) | 24.7 (3.0) | 24.4 (2.3) | 23.8 (2.3) |

* The number of individuals that have withdrawn from each group

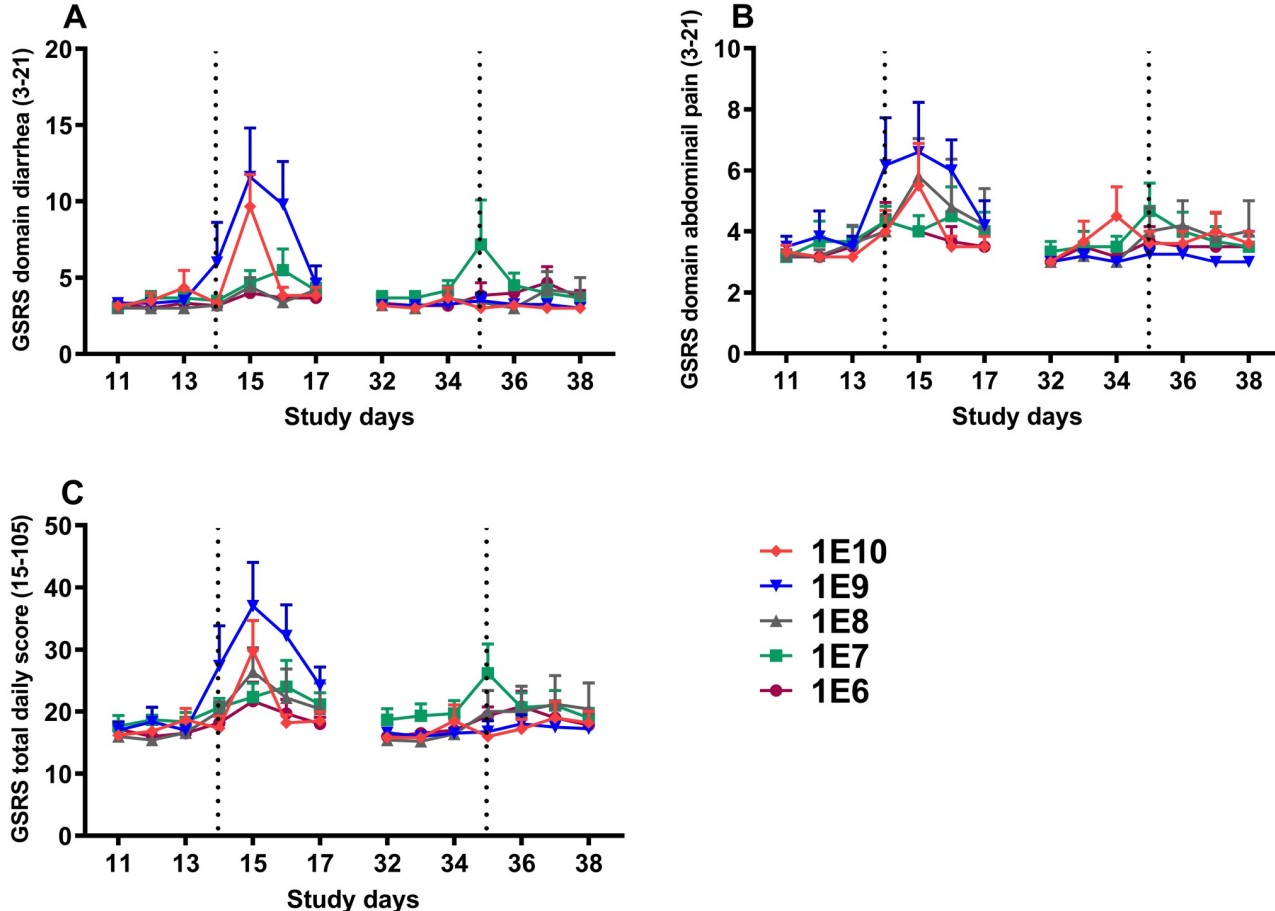

**Fig 3. GSRS domain diarrhea (A), GSRS domain abdominal pain (B), and GSRS total daily score (C) before and after dia.** *E. coli* challenges. The increase in GSRS domain diarrhea score was significantly dose-dependent (P<0.05) after the first challenge but not after the second one. Following the primary challenge, both GSRS domain abdominal pain and GSRS total daily score also increased, without dose-dependency, where an increase was not noted after the second challenge. The dotted lines represent challenge days 14 and 35 of the study, and each symbol represents the mean and one-sided SEM of the group. Data were analyzed using repeated-measures Generalized Estimating Equations (GEE) model.

demonstrated a non-significant (p-value = 0.07) dose-related increasing trend in interval I (Fig 4C) and a significant (p-value <0.05) decreasing linear trend during interval II (Fig 4D).

Moreover, following the primary dia. *E. coli* inoculation, an increase in the frequency of defecations (stools per day) of the participants with a significant dose-dependency (p<0.05) was recorded (Fig 4E). In contrast, in interval II, the challenge dose was inversely related to the stool frequency (p<0.05), indicating a lower number of stools per day in participants of the groups which initially received higher doses of bacteria. Notably, all stool frequency scores in interval II were below the level of clinical diarrhea. Besides, there was no significant relationship between the primary bacteria doses and the stools' consistency based on Bristol Stool Score (BSS) in either of the intervals. This was true for both the average (Fig 4F) and maximum per day stool consistency score (S2 Fig).

## Intestinal immune biomarkers

Three local intestinal immune markers: the inflammatory marker calprotectin, the antimicrobial peptide β-defensin, and total SIgA were analyzed in fecal water extracted from participants' stool samples collected in interval I (days 12/13-17) and interval II (days 33/34-38).

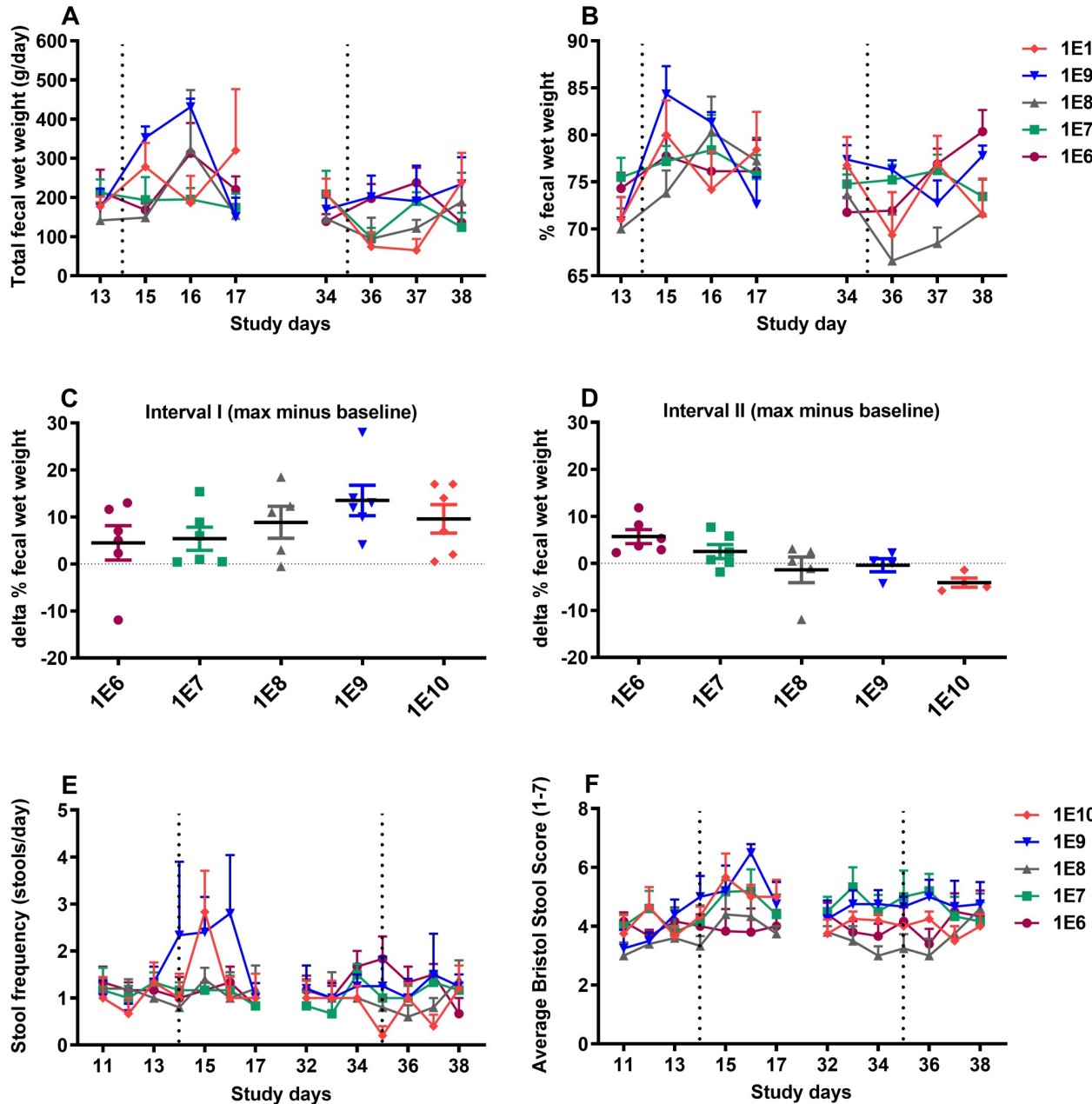

**Fig 4. Total fecal wet weight (A), % of fecal wet weight (%WW) (B), delta % fecal wet weight (Δ%WW) in interval I (C), delta % fecal wet weight (Δ%WW) in interval II (D), stool frequency (E), and stool consistency (BSS) (F), before and after dia.** *E. coli* challenges. The changes in total fecal wet weight, % fecal wet weight, and average Bristol Stool Score (BSS) were not dose-dependent after primary or secondary challenges. The stool frequency was positively related to the primary dia. *E. coli* dose (p<0.05) and negatively related to the dose after rechallenging (p<0.05). The Δ%WW increased relative to the primary dia. *E. coli* challenge dose in interval I and decreased in interval II; checking the linear trend using the one-way ANOVA post-test, showed significance for interval II (p<0.05) but not interval I. The dotted lines represent challenges on day 14 and day 35 of the study, and each symbol represents the mean and SEM of the group. Data (A, B, E, and F) were analyzed using repeated-measures Generalized Estimating Equations (GEE) model and one-way ANOVA post-test was applied to check the linear trend the data in (C and D).

During interval I, a quick rise in calprotectin release was observed for the highest doses (1E10 and 1E9, and 1E8) compared to a delayed response in two lower doses (Fig 5A). Still, all primary dia. *E. coli* inoculation doses reached comparable levels by day 17. The response pattern was different in interval II, with no substantial increase in calprotectin levels. At

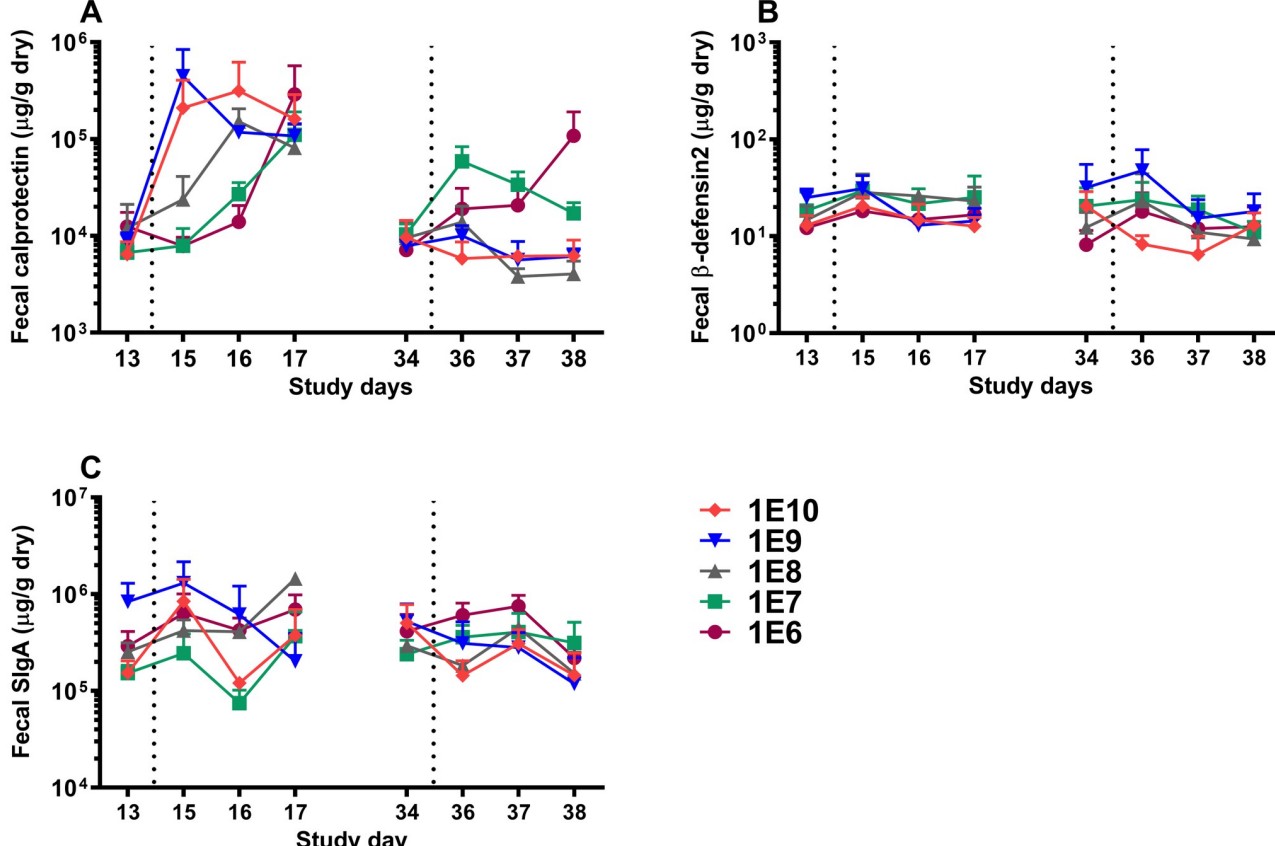

**Fig 5. Fecal calprotectin (A), fecal β-defensin2 (B), and fecal total SIgA (C).** Calprotectin production started on day 15 for doses 1E10 and 1E9, whereas for other doses, it started two days after the primary challenge (day 16). All doses reached similar levels on day 17 (A), and the levels returned to original status just before the second challenge and showed no major variations for any of the primary dose groups after the second challenge. β-defensin (B) and SIgA (C) production were not affected by various dia. *E. coli* doses during the study. The dotted lines represent challenges on day 14 and day 35 of the study, and each symbol represents the mean and one-sided SEM of the group. Data were analyzed using repeated-measures Generalized Estimating Equations (GEE) model.

rechallenge, only participants primarily inoculated with 1E6 and, to a lesser extent, 1E7 CFU showed a slight increase in calprotectin production, and the participants receiving higher primary doses were not affected. The primary infectious dose and the level of β-defensin did not show any relations in either of the intervals (Fig 5B), and generally, only minor changes in β-defensin production were recorded throughout the study. Likewise, fecal SIgA levels were not significantly correlated with the primary challenge dose (Fig 5C).

## Serum levels of IgG-CFA/II

Serum levels of IgG-CFA/II were measured at baseline and weekly for three weeks after each challenge as the study's primary immune parameter. Statistical analysis was performed to compare the absolute IgG levels for different doses over time during the study period (Fig 6).

The low baseline IgG-CFA/II antibody levels increased significantly (p<0.05) for all primary dia. *E. coli* doses throughout the study period. Moreover, different primary challenge doses had an overall significant (p<0.05) positive contribution to the IgG levels meaning the higher the dose, the higher the IgG level. This is illustrated by a positive association between primary dia. *E. coli* doses introduced at day 14 and IgG levels measured at day 35. Noticeably,

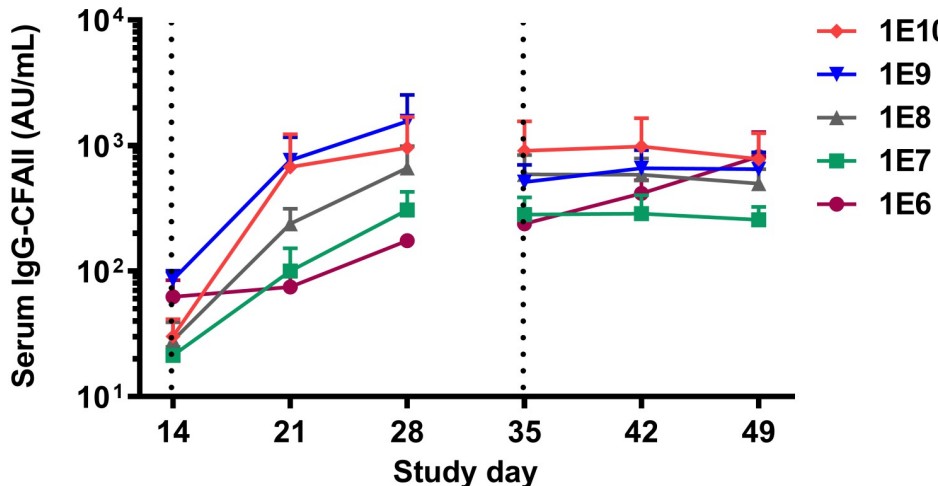

**Fig 6. Serum IgG-CFA/II changes after the first and second challenges with dia. *E. coli*.** The antibody levels increased significantly in all the groups and this increase was significantly dose-dependent. There was no significant increase in serum antibody levels after the second challenge in any doses except for 1E6. The dotted lines represent challenges on day 14 and day 35 of the study, and each symbol represents the mean and one-sided SEM of the group. Data were analyzed using repeated-measures Generalized Estimating Equations (GEE) model.

the IgG-CFA/II levels did not increase further following the second challenge, except for the lowest dose group (1E6 CFU).

### *Ex vivo* stimulation of monocytes, mDC, and pDC by dia. *E. coli* and TLR ligands

To address the innate immune system's capability to respond to *E. coli*, PBMCs isolated from the whole blood collected from the participants just before the challenge on day 14 and day 35 were stimulated with LPS (TLR4 ligand), flagellin (TLR5 ligand), or whole dia. *E. coli*. The percentage of IL-6 and TNF-α positive monocytes, myeloid dendritic cells (mDCs), and plasmacytoid dendritic cells (pDCs) was measured after a 3-hour *ex vivo* stimulation of the cells. These data are expressed as % of double-positive cells (producing both cytokines) or as % of all cells producing IL-6 or TNF-α (single positive). The data were analyzed separately for the different primary doses (S3 and S4 Figs) and for participants of each challenge combined (Fig 7).

Stimulation of PBMCs with LPS, flagellin, and whole dia. *E. coli* bacteria resulted in a higher percentage of double-positive monocytes measured at day 35 compared with day 14 (Fig 7A). The background levels (RPMI medium group) were low on both day 14 and day 35, assuring the observed variations' validity. The same response pattern was observed for all different primary challenge doses of dia. *E. coli* separately (S3 Fig) and when all the doses were combined (Fig 7A). In fact, the average percentage of monocytes producing both IL-6 and TNF-α in response to dia. *E. coli* almost doubled from day 14 to day 35, and nearly all individuals showed an enhancement in response (Fig 7B). Following the monocyte's response pattern, stimulation of PBMCs with LPS, flagellin, and dia. *E. coli*, resulted in a significant increase in the percentage of double-positive mDCs. The observed response pattern was present for all dia. *E. coli* primary doses (S4 Fig) and was significant when all primary doses were combined (Fig 7C). Nearly all individuals had an increase in double-positive mDCs upon dia. *E. coli* stimulation, with an average percentage increase of 1.5 times (Fig 7D). Analysis of IL-6 or TNF-α single-positive cells confirmed the pattern observed for double-positive cells (S5 Fig). pDCs did not

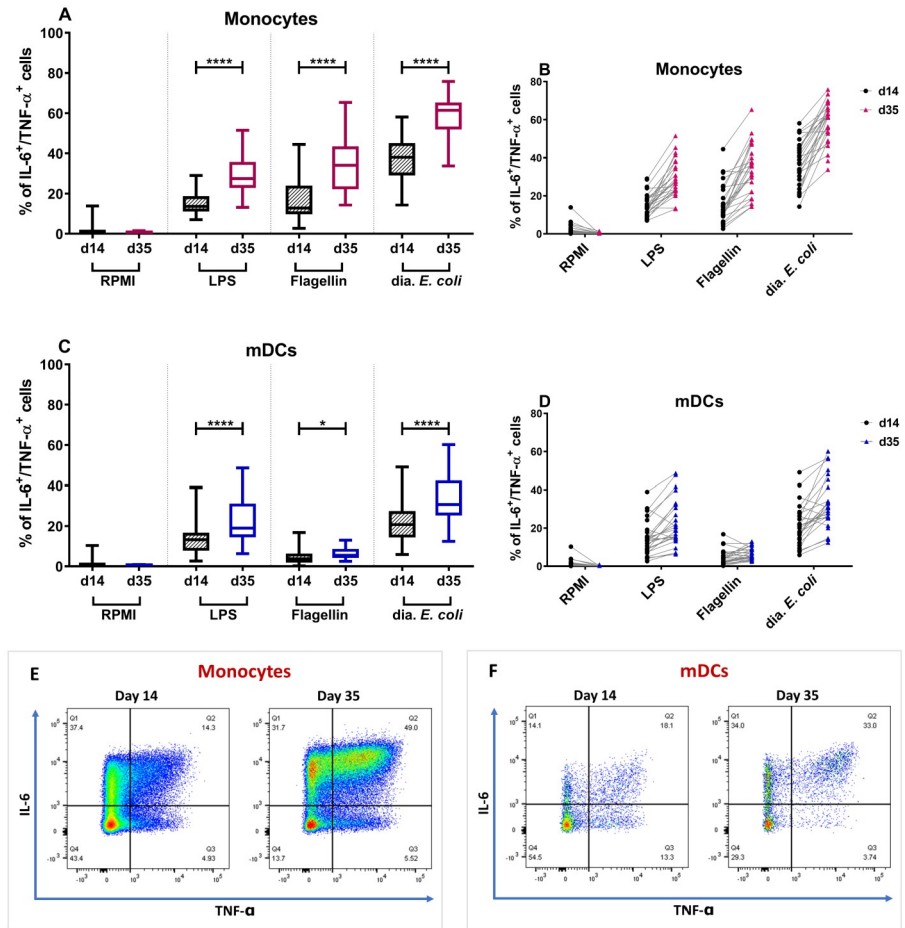

**Fig 7. Variations in double-positive (IL-6 and TNF-α positive) monocytes and mDCs after dia.** *E. coli* **and TLR stimulation on days 14 and 35.** A significant increase in the percentage of monocytes that were simultaneously producing IL-6 and TNF-α was seen after *ex vivo* stimulation with LPS (200 ng/mL), flagellin (500 ng/mL), or dia. *E. coli* (1E7 CFU/well) on day 35, comparing to day 14 (A). The average percentage of double-positive monocytes was almost doubled after stimulation with all stimuli in contrast to the RPMI control (B). Following the stimulation with the same stimuli, mDCs showed a significant increase in IL-6 and TNF-α production after exposure to LPS, flagellin, and dia. *E. coli* (C). Just about all participants showed an increase in double-positive mDCs after dia. *E. coli* and LPS stimulation, with an average percentage increase of almost 1.5 times (D). For statistical analysis, paired sample t-test was applied. Data are represented in Whisker-plots with a median, 25% and 75% quartile. * p<0.05; ** p<0.01; *** p<0.001,**** p<0.0001. A visual representation of changes in the percentage of IL-6 and TNF-α producing monocytes (E) and mDCs (F) of a representative participant (#10) on day 14 compared to day 35. The percentage of double-positive (top-right quadrant) monocytes increased from 14.3% to 49% (E), whereas double-positive mDCs had an increase from 18.1% on day 14 to 33% on day 35 (F) in this selected participant.

respond strongly to any of the stimuli, and changes in the percentage of double-positive pDCs were not significant (S6 Fig).

## Correlation analysis

Statistical analysis was performed to analyze whether serum IgG-CFA/II level before the second challenge (day 35) was correlated with protection against clinical symptom after the challenge. The analysis included the correlation between IgG-CFA/II levels and the following parameters: total fecal weight, % fecal wet weight, stool consistency, stool frequency, GSRS total, GSRS diarrhea, and GSRS abdominal pain. The analyses were based on using all

individual data absolute values of IgG-CFA/II level on day 35 in relation to clinical symptoms on days 35–38. Overall, the analysis results suggest no statistically significant correlation between IgG-CFA/II levels and any clinical symptoms. Following the secondary challenge, a similar approach was applied to check the correlation between the monocytes and mDCs double-positive cells with the clinical outcomes.

## Discussion

Here we demonstrate that exposing healthy volunteers to diarrhoeagenic *E. coli* strain E1392/75-2A dosages as low as 1E6 CFU during a primary infection challenge resulted in clinical protection against a second challenge with a high dose of the same bacteria. This protection was accompanied by increased serum anti-CFA/II IgG levels and enhanced monocyte and mDC responses to *ex vivo* stimulation. However, these enhanced innate and adaptive immune parameters did not correlate significantly with protection to rechallenge.

Well-established and standardized oral infection challenge models are needed to demonstrate the clinical and immunological effects of drug and dietary interventions in preventing infections. Several ETEC challenge models have been described and applied for this reason [16, 20]. In this regard, supplementation with dietary calcium and milk-fat-globule membrane (MFGM) was shown to improve *in vivo* resistance to dia. *E. coli* strain E1392/75-2A -the same strain used in the current study- and Bismuth Subsalicylate ingestion reduced the diarrhea incidence caused by ETEC strain H10407 [9, 12, 14]. While these models aim to prevent primary infection by enhancing passive immunity, we aimed to study correlates of protection to reinfection with the same pathogen using a two-tier dia. *E. coli* challenge model. In this model, varying primary challenge doses of dia. *E. coli* strain E1392/75-2A is followed by a secondary challenge with a high dose of the same bacteria. The *E. coli* strain E1392/75-2A is a well-characterized live-attenuated variant of the ETEC strain O6:H16 that has a spontaneous deletion of enterotoxin-encoding genes but continues to produce CFA/II. Despite the induction of mild and transient gastrointestinal symptoms upon primary challenge, which makes it acceptable to use in human challenge studies, the protection against this strain was evidenced to provide 75% protection (scored as reduction of diarrhea incidence) against wild-type enterotoxin producing strains [16, 21].

Recently, van Hoffen et al. demonstrated that a primary challenge with standard high dose (1E10 CFU) of dia. *E. coli* strain E1392/75-2A largely protected the participants from reinfection with the same pathogen [15]. The protection was accompanied by an increased serum IgG-CFA/II response. A higher primary dose of 5E10 CFU, on the other hand, did not affect the severity of clinical symptoms or the anti-CFA/II antibody response following the second challenge. Association analyses could not confirm a direct correlation between serum IgG-CFA/II titers and the intensity of clinical symptoms, and therefore, it could not be concluded that IgG-CFA/II serum levels provided protection against reinfection.

Because dia. *E. coli* primary challenge doses of 1E10 CFU or higher conferred total protection, identifying immune parameters that contributed to the protection against reinfection was not possible. The current study was designed as a dose-response pilot study to identify the underlying immunological mechanisms linked to correlates of protection. After the primary challenge, the lowest two doses of dia. *E. coli* did not induce clinical symptoms but increasing the bacteria dose resulted in clinical symptoms. This was illustrated by a significant dose-dependent increase in stool frequency and GSRS score in the subdomain diarrhea (Figs 4E and 3A). These parameters serve as primary indicators of diarrhea occurrence and were reported to increase following the inoculation of participants with high doses of dia. *E. coli* E1392/75-2A [14, 15]. Similarly, GSRS domain abdominal pain and GSRS daily total score

presented a consistent but not significant trend towards increasing for the two highest doses. This is apparent from the kinetic pattern of the responses (Fig 3B and 3C). Moreover, when focusing on the variations in Δ%WW -as a clear indicator of changes in fecal wet weight related to diarrhea- it is quite illustrative that the challenge dose directs the severity of diarrhea after primary challenge (Fig 4C). Interestingly, there is a putative trend that higher doses protect better against the clinical symptoms, although all tested doses in primary challenge confer protection at rechallenge (Fig 4D).

After primary or secondary challenges, the analysis of β-defensin and total SIgA levels in fecal water showed no significant dia. *E. coli*-induced dose-dependent protein production (Fig 5B and 5C). However, the highest two doses of 1E10 and 1E9 significantly induced fecal calprotectin on day 15 (Fig 5A), the first day after the primary challenge, which is in line with previous reports [14, 18]. Calprotectin is a calcium-binding protein serving as a key marker of acute intestinal inflammation resulting from the influx of leukocytes, mainly neutrophils, into the gut as a response to infection [22]. Calprotectin induction in the fecal extract of participants exposed to low levels of dia. *E. coli* (1E6 and 1E7 CFU), and to a lesser extent, in those exposed to 1E8 CFU, did not occur until day 17, three days after the primary challenge. (Fig 5A). It is known that dia. *E. coli* strain E1392/75-2A only temporarily colonizes the gut and is typically cleared within 14 days after challenge with 1E10 CFU [9]. A possible explanation for the delayed dose-dependent induction of calprotectin at low infectious dosages observed here may be that the low level of infection allows a temporary expansion of the bacteria, reaching a certain threshold level to result in calprotectin production before the infection is cleared.

A dose-dependent increase of CFA/II-specific IgG levels in the serum was noted on day 28 of the study (Fig 6). This was especially obvious when comparing the two highest doses of 1E9 and 1E10 to the lower doses. The observed dose-response pattern in CFA/II-specific IgG levels was consistent with previous findings reviewed by Porter et al., indicating that lower inoculation doses may lead to lower antibody response after primary challenge [20].

Contrary to our expectations, rechallenging with a high dose of dia. *E. coli* 1E10 CFU did not result in severe clinical symptoms in any of the dose groups tested. Even at the lowest dose (1E6 CFU), there were almost no reports of severe diarrhea or gastrointestinal symptoms (Fig 3). In fact, after rechallenge, all GSRS scores and stool parameters, including total and relative fecal wet weight, as well as stool frequency and consistency markers, showed minor fluctuations, indicating that all participants were protected from diarrhea regardless of the primary dia. *E. coli* challenge dose. Likewise, no intestinal inflammation was detected, as evidenced by the absence of an apparent increase in calprotectin levels in the subjects' fecal extracts (Fig 5A). After the second challenge, only those who received the lowest primary dia. *E. coli* doses of 1E6 and 1E7 experienced a slight increase in calprotectin levels. A possible explanation for the low inflammatory responses could be the exclusion of dia. *E. coli* by secretory IgA in the gut and, consequently, less interaction of the bacteria with mucosal epithelial receptors [23, 24].

Upon rechallenge, serum IgG-CFA/II level hardly increased further in any of the groups (Fig 6), suggesting attainment to a steady-state what was reported before for dose 1E10 of the same dia. *E. coli* strain [15] and for ETEC strain H10407 [25]. In our study, only two patients from the dose group 1E6 who did not respond to the primary challenge showed an increase in IgG-CFA/II levels following the second challenge. This finding suggests that, like calprotectin, a minimal inoculation dose is required to induce an IgG-CFA/II response, which may vary between participants. Later association analyses could not reveal a significant correlation between serum IgG-CFA/II level on day 35 and any clinical symptoms after the second challenge, similar to previous observations. To our knowledge, although systemic and local parameters have been measured in dia. *E. coli* challenge studies, no direct correlation between serum

IgG/IgA or fecal IgA titer and clinical protection has been documented yet [20, 26]. Despite no documented correlations, the role of pathogen-specific IgG and IgA and, most importantly, SIgA -which facilitates bacteria clearance from the gut- in establishing protection against dia. *E. coli* may not be disregarded and needs to be addressed in future studies.

The relative relevance of the innate immune system in protection against dia. *E. coli* reinfection has not been previously addressed. As two key innate immune cells, monocytes and mDCs are among the first cells to respond to pathogen translocation across the epithelial barrier by releasing pro-inflammatory mediators such as IL-6 and TNF-α. Their cytokine secretion profile and antigen presentation to naïve lymphocytes are critical for inducing host defense mechanisms and regulating both innate and adaptive immune responses [27, 28]. Interestingly, we observed an increased response of monocytes and mDCs to dia. *E. coli* -as well as to LPS and flagellin- three weeks after the primary challenge, regardless of the initial challenge dose used (Fig 7, S3 and S4 Figs). Given a 3-week period between the primary challenge and the increased monocyte and mDC response, a priming effect on these cells by the first challenge does not seem likely. An alternative explanation can be that the primary challenge with dia. *E. coli* leads to trained immunity.

The concept of trained immunity is based on observations that individuals who received the Bacillus Calmette–Guérin (BCG) vaccine develop cross-protection against unrelated pathogens [29, 30]. Netea and colleagues verified the findings and not only demonstrated the training ability of some other microbial components such as β-glucan but also described epigenetic modifications, e.g., increased H3K4 trimethylation as the underlying mechanism [31–34]. Monocytes undergo an inheritable alteration in their gene expression pattern following training by potential components [35, 36]. After training, these cells exhibit an enhanced nonspecific response upon secondary TLR-mediated activation by the same and similar pathogens, as evidenced by increased pro-inflammatory cytokine production. While trained immunity was initially demonstrated in monocytes and later natural killer (NK) cells, new research indicates that other myeloid cells, including dendritic cells, may also be trained to elicit an increased response to repeated stimulation [37]. Therefore, the increased number of activated monocytes and mDCs after the primary challenge presented in this study suggests that dia. *E. coli* strain E1392/75-2A may induce trained immunity *in vivo*.

This hypothesis is supported by the fact that the enhanced responses were not limited to the applied dia. *E. coli* strain. Apart from the applied strain, the augmented number of IL-6 and TNF-α producing monocytes and mDCs were recorded upon the stimulation with TLR ligands, namely LPS (from *E. coli* strain O55:B5) and flagellin (from *Salmonella typhimurium*). This indicates that the enhanced innate response to the dia. *E. coli* is also extended towards unrelated pathogens. To which extent and for how long these effects will last is currently unknown.

This study links local and systemic immune parameters to clinical gastrointestinal symptoms following dia. *E. coli* infection to explain the immunological factors related to clinical protection against reinfection in this challenge mode. Even though no dose-dependent correlation between immune markers and clinical symptoms was noted, we speculate that the observed protection may have resulted from the participation of multiple immunological components. Both humoral immunity -mainly via serum anti-CFA/II IgG and SIgA- and innate immunity -via the enhanced activity of monocytes and mDCs- may play a role in the clinical protection against infection.

In conclusion, our data suggest that low-dose dia. *E. coli* exposure can protect against reinfection through modulating adaptive and innate immune responses. The lower IgG and calprotectin responses induced by primary challenge, as well as the trend in % fecal wet weight at rechallenge, may be used to further establish this two-tier dia. *E. coli* challenge model as a model to explore the impact of drugs and food components on antibacterial immunity.

## Supporting information

**S1 Fig. Gating strategy for selecting monocytes, mDCs, and pDCs.** In the FSC/SSC plot cells within the PBMC region were selected. The duplets were gated out and the HLA-DR$^+$CD14$^+$ cells were considered as the monocytes. From the HLA-DR$^+$CD14$^-$ population CD3$^+$, CD19$^+$, CD20$^+$ and CD56$^+$ cells were excluded. In the remaining population, CD11c$^+$ cells were considered as mDCs and CD123$^+$ cells were named pDCs. Within monocyte, mDC, and pDC populations cells that were producing IL-6, TNF-α or both cytokines were determined.
(TIF)

**S2 Fig. Maximum daily Bristol Stool score.** The changes in maximum Bristol stool score (stool consistency score) after primary infection or after reinfection were not dependent on the bacterial dose used during primary infection. The dotted lines represent infection days 14 and 35 of the study, and each symbol represents the mean and one-sided SEM of the group. Data were analyzed using repeated-measures Generalized Estimating Equations (GEE) model.
(TIF)

**S3 Fig. Variation in the percentage of double-positive monocytes within different primary infection group doses.** After *ex vivo* stimulation of PBMCs with either medium (RPMI-1640), 200 ng/mL of LPS, 500 ng/mL of Flagellin, or 1E7 CFU/well of *E. coli* (strain E1392/75-2A), the percentage of double-positive monocytes increased in all dose groups. This increase was significant after stimulation with flagellin and *E. coli* in group dose 1E6 and 1E7, flagellin in group dose 1E8, and following LPS, flagellin, ETEC stimulation in group 1E10. Data are shown in Whisker-plots with a median, 25% and 75% quartile. $^*$ p<0.05; $^{**}$ p<0.01; $^{***}$ p<0.001.
(TIF)

**S4 Fig. Variation in the percentage of double-positive mDCs within different primary infection group doses.** After *ex vivo* stimulation of PBMCs with either medium (RPMI-1640), 200 ng/mL of LPS, 500 ng/mL of Flagellin, or 1E7 CFU/well of E. coli (strain E1392/75-2A), the percentage of double-positive mDCs increased in all dose groups. This increase was significant after stimulation with E. coli in group dose 1E7. Data are shown in Whisker-plots with a median, 25% and 75% quartile. $^*$ p<0.05; $^{**}$ p<0.01; $^{***}$ p<0.001.
(TIF)

**S5 Fig. The variation in the percentage of all the cells that were producing IL-6 or TNF-α (single positives) after *ex vivo* stimulation.** The cells were stimulated *ex vivo* with either medium (RPMI-1640), 200 ng/mL of LPS, 500 ng/mL of Flagellin, or 1E7 CFU/well of *E. coli* (strain E1392/75-2A). The percentage of IL-6$^+$ as well as TNF-α$^+$ monocytes increased significantly on day 35 compared to day 14. In mDCs, only LPS and *E. coli* increased the percentage of IL-6$^+$ cells on day 35 and the significant increase in TNF-α$^+$ mDCs only occurred after ETEC stimulation on day 35. Data are shown in Whisker-plots with a median, 25% and 75% quartile. $^*$ p<0.05; $^{**}$ p<0.01; $^{***}$ p<0.001.
(TIF)

**S6 Fig. Variations in double-positive (IL-6 and TNF-α positive) pDCs after ETEC and TLR stimulation on days 14 and 35.** *Ex vivo* stimulation of pDCs with either medium (RPMI-1640), 200 ng/mL of LPS, 500 ng/mL of Flagellin, or 1E7 CFU/well of E. coli (strain E1392/75-2A), did not result in an increased percentage of double-positive pDCs. The Amount of IL-6 and TNF-α production in pDCs were marginal in all challenge doses and upon all different stimuli and were not significantly different between base line and day 21 of the study.
(TIF)

**S1 Checklist.**
(PDF)

**S1 File.**
(PDF)

## Author Contributions

**Conceptualization:** Mojtaba Porbahaie, Laurien Ulfman, Rianne M. A. J. Ruijschop, Wim Calame, Els van Hoffen, R. J. Joost van Neerven, Alwine Kardinaal.

**Data curation:** Mojtaba Porbahaie.

**Formal analysis:** Mojtaba Porbahaie, Wim Calame.

**Investigation:** Mojtaba Porbahaie, Maartje van den Belt, Elly Lucas–van de Bos, Anita Hartog, Stefanie Lenz, Ingrid J. van Alen-Boerrigter.

**Methodology:** Mojtaba Porbahaie, Maartje van den Belt, Elly Lucas–van de Bos, Stefanie Lenz, Ingrid J. van Alen-Boerrigter.

**Project administration:** Mojtaba Porbahaie, Maartje van den Belt, Rianne M. A. J. Ruijschop, Anita Hartog, Els van Hoffen, Alwine Kardinaal.

**Supervision:** Laurien Ulfman, Rianne M. A. J. Ruijschop, Els van Hoffen, R. J. Joost van Neerven, Alwine Kardinaal.

**Validation:** Laurien Ulfman, Anita Hartog, Els van Hoffen, R. J. Joost van Neerven, Alwine Kardinaal.

**Writing – original draft:** Mojtaba Porbahaie.

**Writing – review & editing:** Mojtaba Porbahaie, Maartje van den Belt, Laurien Ulfman, Rianne M. A. J. Ruijschop, Elly Lucas–van de Bos, Anita Hartog, Stefanie Lenz, Ingrid J. van Alen-Boerrigter, Malgorzata Teodorowicz, Huub F. J. Savelkoul, Wim Calame, Els van Hoffen, R. J. Joost van Neerven, Alwine Kardinaal.

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
