## [Decision Letter · Decision Letter 0]

15 Aug 2022

PONE-D-22-00136Low doses of diarrhoeagenic E. coli induce enhanced monocyte and mDC responses and prevent development of symptoms after homologous rechallengePLOS ONE

Dear Dr. Porbahaie,

Thank you for submitting your manuscript to PLOS ONE. After careful consideration, we feel that it has merit but does not fully meet PLOS ONE’s publication criteria as it currently stands. Therefore, we invite you to submit a revised version of the manuscript that addresses the points raised during the review process.

Specifically, the manuscript was evaluated by two reviewers and they raised a couple concerns including sample calculation and sample selection. Please provide your responses to all the comments point-by-point.

We look forward to receiving your revised manuscript.

Kind regards,

Jianhong Zhou

Staff Editor

PLOS ONE

Journal Requirements:

“ The CHALLENGE consortium is financially supported by Operationeel Programma Oost-Nederland (Operational Program Eastern Netherlands), a joint program of the provinces of Overijssel and Gelderland, the city networks Zwolle, Kampen, Apeldoorn, Deventer, Zutphen, and Arnhem-Nijmegen, and the regional networks in Twente and The Valley. The program covers activities that are co-financed by the European Regional Development Fund.”

3. Thank you for stating the following in the Competing Interestssection:

“L. Ulfman and R.J.J. van Neerven are employed by FrieslandCampina. Other authors declare that there is no conflict of interest.”

We note that one or more of the authors are employed by a commercial company: FrieslandCampina

Reviewers' comments:

Reviewer's Responses to Questions

**Comments to the Author**

1. Is the manuscript technically sound, and do the data support the conclusions?

Reviewer #1: Partly

Reviewer #2: Yes

2. Has the statistical analysis been performed appropriately and rigorously? 

Reviewer #1: No

Reviewer #2: I Don't Know

3. Have the authors made all data underlying the findings in their manuscript fully available?

Reviewer #1: No

Reviewer #2: Yes

4. Is the manuscript presented in an intelligible fashion and written in standard English?

Reviewer #1: Yes

Reviewer #2: Yes

5. Review Comments to the Author

Reviewer #1: This manuscript reports a dose-response pilot study investigating which immunological factors are associated with this protection using healthy subjects. I have below comments and questions.

In Power calculation, it’s not clear how n=6 was calculated. How big power was designed to achieve for what outcome or hypothesis test in this study? For this pilot study, power calculation is not necessary. Why do you use Plackett-Burman design? How did you use it? It is not related to this dose-response study. The cited reference 18 has nothing about Plackett-Burman design.

In statistical analysis, it is not clear what stepwise approach was used and why it as needed. What nonlinear function of day and dose were included in GEE analysis? What software was used? Does “Fit of the model” mean “goodness of fit for the model” in your statement? What do you mean “when fit was significant”?

Fig 4C and D may be analyzed using Cuzick’s test for trend.

In Correlation analysis, what statistical method was used for correlation?

In each figure legend, please add what statistical methods were used for analysis.

Reviewer #2: In their manuscript entitled: “Low doses of diarrheagenic E. coli induce enhanced monocyte and mDC responses and prevent development of symptoms after homologous rechallenge, the authors describe a challenge-rechallenge study in human volunteers to assess the protective immunity developed from low vs. high dose challenge of a diarrheagenic E. coli strain. They found that the symptoms and diarrhea induced by the primary challenge to be correlated with the challenge dose, all doses protected the volunteers from symptoms upon rechallenge, with stool frequency decreased in those who received the higher primary dose. Immunological markers of inflammation were greater during the primary challenge, with less inflammation during the secondary challenge, with the exception of dual IL-6 and TNF-alpha secreting PBMCs and monocytes, which increased after the second challenge. The manuscript was well written and easy to read.

I just have a few comments:

Some of the material in the methods section (Figure 1) is better in the results section.

Why were only males included in the study? Why were females excluded?

Minor comment:

Methods, p 13 line 133: if the dietary calcium intake was restricted, was it limited to <500 mg/day? Or is it correct as written?

6. PLOS authors have the option to publish the peer review history of their article (what does this mean?). If published, this will include your full peer review and any attached files.

Reviewer #1: No

Reviewer #2: No

---

## [Author Response · Author response to Decision Letter 0]

7 Oct 2022

Dear Jianhong Zhou,

An updated cover letter containing new "funding statement" and " competing interest statement" was uploaded. A separate file was uploaded as the response to the reviewers. A clean version of the manuscript and a version with track changes were uploaded. Please contact me if further information is needed.

Regards,

Mojtaba Porbahaie

---

## [Decision Letter · Decision Letter 1]

21 Nov 2022

PONE-D-22-00136R1Low doses of diarrhoeagenic E. coli induce enhanced monocyte and mDC responses and prevent development of symptoms after homologous rechallengePLOS ONE

Dear Dr. Porbahaie,

Thank you for submitting your manuscript to PLOS ONE. After careful consideration, we feel that it has merit but does not fully meet PLOS ONE’s publication criteria as it currently stands. Therefore, we invite you to submit a revised version of the manuscript that addresses the points raised during the review process.

Specifically, please address the remaining comments raised the reviewers.

We look forward to receiving your revised manuscript.

Kind regards,

Jianhong Zhou

Staff Editor

PLOS ONE

Journal Requirements:

Reviewers' comments:

Reviewer's Responses to Questions

**Comments to the Author**

1. If the authors have adequately addressed your comments raised in a previous round of review and you feel that this manuscript is now acceptable for publication, you may indicate that here to bypass the “Comments to the Author” section, enter your conflict of interest statement in the “Confidential to Editor” section, and submit your "Accept" recommendation.

Reviewer #1: (No Response)

Reviewer #2: All comments have been addressed

2. Is the manuscript technically sound, and do the data support the conclusions?

Reviewer #1: (No Response)

Reviewer #2: Yes

3. Has the statistical analysis been performed appropriately and rigorously? 

Reviewer #1: (No Response)

Reviewer #2: I Don't Know

4. Have the authors made all data underlying the findings in their manuscript fully available?

Reviewer #1: (No Response)

Reviewer #2: Yes

5. Is the manuscript presented in an intelligible fashion and written in standard English?

Reviewer #1: (No Response)

Reviewer #2: Yes

6. Review Comments to the Author

Reviewer #1: Please note, the reference list needs update. The reference 18 need to be replaced to Plackett-Burman’s paper.

Reviewer #2: In their revised paper entitled “Low doses of diarrhoeagenic E. coli induce enhanced monocyte and mDC responses and prevent development of symptoms after homologous rechallenge” the authors have addressed the queries of the reviewer.

This reviewer is comfortable with the changes to the paper. I would remove the sentences explaining why this study does not include women. The reasoning of the authors is not consistent with other challenge studies conducted- there has been no difficulty collecting stool separately from women despite their anatomy, and most women can distinguish abdominal pain from a diarrheagenic infection vs. menstrual cramps.

Minor comment:

Results, Gastrointestinal symptom rating scale:

Line 261: “neither” should be “either”

7. PLOS authors have the option to publish the peer review history of their article (what does this mean?). If published, this will include your full peer review and any attached files.

Reviewer #1: No

Reviewer #2: No

---

## [Author Response · Author response to Decision Letter 1]

24 Nov 2022

Response to reviewers: 

Reviewer #1: Please note, the reference list needs update. The reference 18 need to be replaced to Plackett-Burman’s paper. 

Reply: Thank you for pointing out this issue. The reference was updated in the revised version and the mentioned reference is now number 19. 

Reviewer #2: In their revised paper entitled “Low doses of diarrhoeagenic E. coli induce enhanced monocyte and mDC responses and prevent development of symptoms after homologous rechallenge” the authors have addressed the queries of the reviewer. This reviewer is comfortable with the changes to the paper. I would remove the sentences explaining why this study does not include women. The reasoning of the authors is not consistent with other challenge studies conducted- there has been no difficulty collecting stool separately from women despite their anatomy, and most women can distinguish abdominal pain from a diarrheagenic infection vs. menstrual cramps. 

Reply: We would like to thank the reviewer for the comment. The statement was not initially there and it was added to the revised version 1.0. Based on the recommendations, we decided to remove the sentence. 

Minor comment: Results, Gastrointestinal symptom rating scale: Line 261: “neither” should be “either” 

Reply: The word was corrected in the revised version of the manuscript.

---

## [Editor Report · Decision Letter 2]

13 Dec 2022

Low doses of diarrhoeagenic E. coli induce enhanced monocyte and mDC responses and prevent development of symptoms after homologous rechallenge

PONE-D-22-00136R2

Dear Dr. Porbahaie,

We’re pleased to inform you that your manuscript has been judged scientifically suitable for publication and will be formally accepted for publication once it meets all outstanding technical requirements.

Kind regards,

Jianhong Zhou

Staff Editor

PLOS ONE
---

## [Editor Report · Acceptance letter]

27 Dec 2022

PONE-D-22-00136R2 

Low doses of diarrhoeagenic *E. coli* induce enhanced monocyte and mDC responses and prevent development of symptoms after homologous rechallenge 

Dear Dr. Porbahaie:

I'm pleased to inform you that your manuscript has been deemed suitable for publication in PLOS ONE. Congratulations! Your manuscript is now with our production department. 

Kind regards, 

on behalf of

Jianhong Zhou 

Staff Editor

PLOS ONE